# All-day fresh water harvesting by microstructured hydrogel membranes

Ye Shi [1✉], Ognjen Ilic[1,2], Harry A. Atwater [1] & Julia R. Greer [1✉]

Solar steam water purification and fog collection are two independent processes that could enable abundant fresh water generation. We developed a hydrogel membrane that contains hierarchical three-dimensional microstructures with high surface area that combines both functions and serves as an all-day fresh water harvester. At night, the hydrogel membrane efficiently captures fog droplets and directionally transports them to a storage vessel. During the daytime, it acts as an interfacial solar steam generator and achieves a high evaporation rate of 3.64 kg m$^{-2}$ h$^{-1}$ under 1 sun enabled by improved thermal/vapor flow management. With a homemade rooftop water harvesting system, this hydrogel membrane can produce fresh water with a daily yield of ~34 L m$^{-2}$ in an outdoor test, which demonstrates its potential for global water scarcity relief.

[1] Division of Engineering and Applied Science, California Institute of Technology, Pasadena, CA, USA. [2] Department of Mechanical Engineering, University of Minnesota, Minneapolis, MN, USA. ✉email: yeshi119@utexas.edu; jrgreer@caltech.edu

Water scarcity is among the most serious global challenges of our time, and significant efforts have been dedicated to harvesting fresh water from alternative sources[1–3]. For example, interfacial solar steam generation utilizes sunlight as energy source to purify saline or contaminated water by directly heating water and driving its evaporation at the water-air interface[4–10]. Its efficiency depends on water transport and thermal management, and various materials have been studied for this application. For example, nanostructured carbon materials have been designed to absorb light and facilitate water transport[11–13], and plasmonic materials[14,15] and ceramics[5] were used to efficiently convert sunlight to thermal energy. Yu et al. demonstrated a polyvinyl alcohol (PVA)/polypyrrole (PPy) hydrogel-based solar steam generator to have a vapor generation rate of $3.2 \, kg \, m^{-2} \, h^{-1}$ enabled by expedited water transport through porous matrix and a reduced water vaporization enthalpy within the polymeric mesh[16,17]. Recently they improved the rate to a high record of $\sim 3.6 \, kg \, m^{-2} \, h^{-1}$ by modifying the hydrogel systems with highly hydratable polymers or light-absorbing fillers[18,19]. All of these solar steam generators have major drawbacks in that they can only work under sufficient solar irradiation, and their output is limited by the solar energy density at the earth's surface and the size of energy consumption required for water evaporation. The merit of these materials will be greatly improved if they can harvest other fresh water resources and continuously produce clean water around the clock.

Fog frequently occurs in the coastal and post-sunset arid areas, and is a source of water that is complementary to solar water purification[20–22]. Fog collection presents a promising and low-cost approach to water harvesting, which has been widely studied and employed[23–25]. Polymer mesh materials are commonly used to capture fog[26–28], but their efficiency is adversely affected by re-entrainment of deposited droplets and clogging of the mesh with pinned droplets[29]. Certain natural structures with distinctive functions have been discovered that avoid these problems and collect fog more efficiently[30–32]. For example, the hierarchically assembled conical structures of Cactus spine are able to continuously harvest fog by driving directional movement of droplets[33]. Several bio-inspired fog collection motifs have been explored in which devices are constructed with metals[34], metal oxides[35] and polymers[36,37], all of which lack light-into-thermal energy conversion ability, thus rendering them incompatible with solar steam generation. Developing a structured material that could support both technologies would provide an avenue for exploiting both water collection mechanisms around the clock and would have a significant impact on global water scarcity relief.

We designed and fabricated a PVA/PPy hydrogel membrane populated with three-dimensional (3D) tree-shaped surface microstructures. Our choice of a hydrogel membrane stems from its ability to serve as an effective interfacial solar steam generator for water purification. Coupled with the excellent processability of hydrogels and their compatibility with advanced manufacturing techniques, these viscoelastic materials are easily shaped into microstructures that can mimic biological systems at relevant length scales to facilitate fog collection. When placed under controlled fog generation conditions, this PVA/PPy gel membrane efficiently captures fog droplets at a rate of $\sim 5.0 \, g \, cm^{-2} \, h^{-1}$ and drives droplet transport while providing directional control. Using experiments and modeling, we also demonstrate that the tree-shaped surface micro-topologies enable amplification of thermal and fluidic management for interfacial solar steam generation by maximizing light absorption efficiency and guiding vapor escape, thus enabling a high solar vapor generation rate of $3.64 \, kg \, m^{-2} \, h^{-1}$ under 1 sun irradiation. In outdoor tests, this device is capable of all-day fresh water harvesting and delivers a daily water collection rate of $\sim 34 \, L \, m^{-2}$.

## Results

**Design of PVA/PPy gel membrane with micro-tree array for bi-functional water collection.** Figure 1a is a schematic of the fresh-water-collecting membrane. At night, the hydrogel membrane is exposed to fog, and the surface microstructures continuously capture fog droplets and transport them to a storage vessel. During the daytime, the hydrogel membrane acts as an interfacial solar steam generator to purify saline or contaminated water.

To develop this unique bifunctional water collection membrane, PVA-based hydrogel was selected as the building material. This material choice stems from its favorable solar steam generation ability, water affinity, and processability. PVA hydrogel provides hierarchically porous pathways within its matrix for efficient water transport and it reduces the evaporation enthalpy of water owing to interactions between its hydroxyl groups and water molecules, thus enabling high-performance solar steam generation[16]. Its hydrophilic nature also favors water capture on its surface. Hydrogel materials are compatible with various processing techniques and can be easily shaped into desired structures.

Though PVA hydrogel captures water efficiently, a smooth membrane surface inhibits its fog collection ability since captured droplets will be pinned on its hydrophilic surface. To enable optimally efficient water collection from fog, the surface structure needs to be modified to continuously remove deposited droplets[38,39]. Cactus spine-inspired conical structures were adopted for this purpose. Water droplets attached to the sides of conical structures experience a Laplace pressure difference, $\Delta P$,[40]

$$\Delta P = \frac{dP}{dz}\bigg|_{\Omega} = -\frac{2\gamma}{(r + R_0)^2} \sin \alpha \qquad (1)$$

where $\Omega$ is droplet volume, $\gamma$ is surface tension, $r$ is the local radius, $R_0$ is the droplet radius and $\alpha$ is the half apex angle. This Laplace pressure difference drives droplets towards the wider base, thus re-exposing the gel surface to more incoming vapor. According to Eq. 1, the apex angle in our design is the smallest possible within the constraints of the fabrication process and mechanical strength of PVA hydrogel to increase $\Delta P$ and causes the droplets to move faster.

To increase the surface area and thus provide a benefit for both fog capture and interfacial solar steam generation, we assembled the gel cones in a hierarchical way by building branched small cones on a cone trunk and then arrayed these tree-like structures into a dense forest on a membrane surface. Light absorption is improved in the gel forest and water droplets collected on branches are able to merge together for quick drainage. The density of these gel trees is also carefully tuned to facilitate the escape of generated vapor during steam generation and water drainage during fog collection.

To realize all-day water collection in natural environments, a floating prototype is built to support hydrogel membranes and store collected water. As shown in Fig. 1b, c, a foldable cover is designed in our floating prototype. During night, it's open and the gel samples can be supported to face the fog flow. During daytime, it's closed and acts as a re-condensation structure.

The micro-tree array structure was designed in CAD software (Supplementary Fig. 1) and fabricated on a PVA/PPy gel membrane using stereolithography 3D Printing, followed by a simple molding method (Fig. 2a). The photomicrographs in Fig. 2b illustrate a typical gel membrane with a projected area

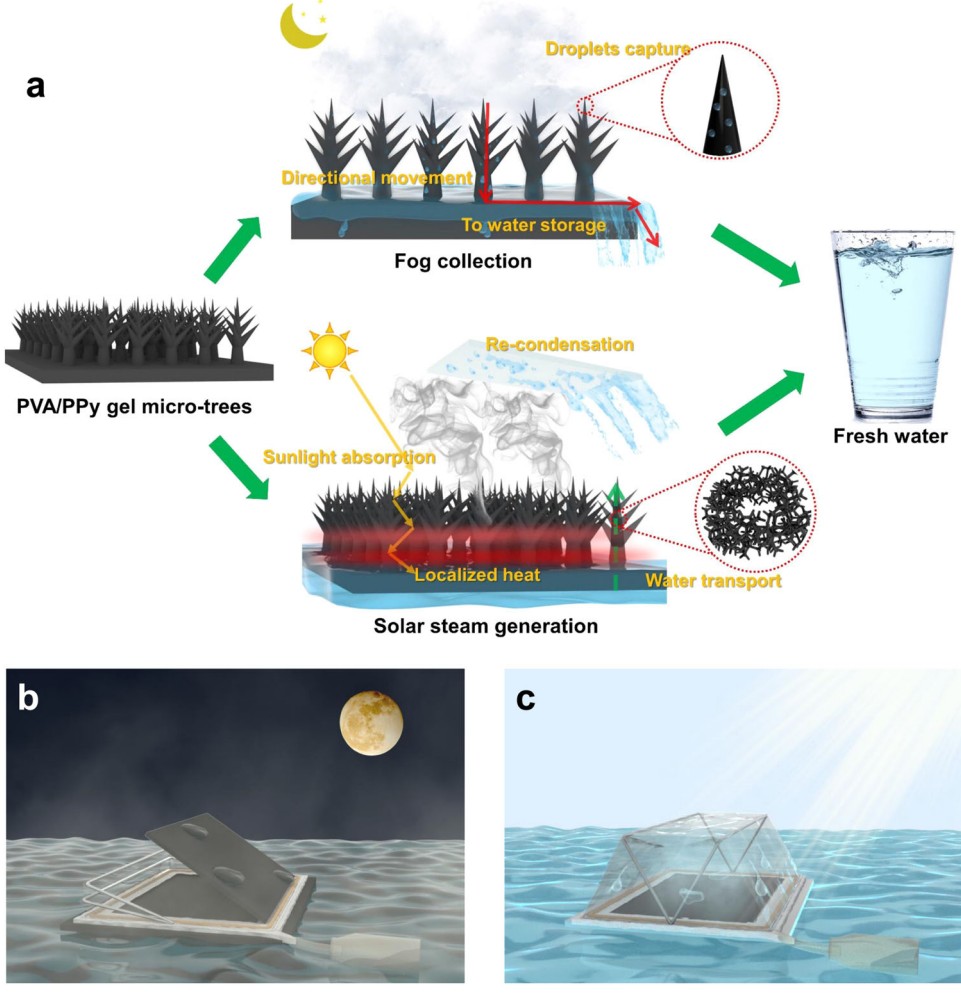

**Fig. 1 Design of the bifunctional gel membranes and all-day water harvesting prototype. a** Conceptual representation of the PVA/PPy hydrogel membrane with micro-topologies that is capable of 24-h fresh water harvesting. **b, c** Schematic illustration of nighttime (**b**) and daytime (**c**) modes of floating device for all-day water harvesting.

($A_p$) of ~5.5 cm$^2$, which contains 100 hexagonally arranged micro-trees on a supporting layer. Each tree is ~4 mm tall, has a bottom diameter of ~0.8 mm, and contains nine 45-degree tilted conical branches at 1/3, 1/2, and 2/3 of the tree height (Fig. 2c). All the branched cones have the same conicity as the trunk. Scanning electron microscope images reveal that the smallest dimension of the conical structure is ~20 μm at the tip (Fig. 2d). The cross-linked hydrogel is hierarchically porous and contains inter-dispersed PPy particles, which may be beneficial in enabling efficient water transportation within the matrix (Fig. 2e and Supplementary Fig. 2)[16]. The chemical composition and mechanical properties of PVA/PPy hydrogels were also investigated by FTIR spectroscopy and rheological measurements (Supplementary Figs. 3, 4a).

**Fog harvesting properties of PVA/PPy gel membrane with micro-tree array**. We conducted fog collection experiments to quantify the water collection rate of microstructured PVA/PPy gels (Supplementary Fig. 5). Our experiments demonstrate that under a continuous fog flow generated by an ultrasonic humidifier, the micro-trees capture micro-sized water droplets that quickly grow and coalesce with one another as they move towards the cone base while new droplets continuously condense onto the cones. As this process continues, droplets from different branches merge together into a millimeter-size droplet, which is ultimately collected into the beaker with the guide of the support layer (Fig. 3a and Supplementary Fig. 6, Supplementary Movie 1). This cycle of fog droplets nucleation followed by their transport, growth, and eventual drainage of the large water drops repeats with an average period of ~20 s, which corresponds to a saturated fog collection rate ($m/A_p$) of ~5.0 g cm$^{-2}$ h$^{-1}$ calculated using the projected membrane area (Fig. 3b). Note that our fully hydrated hydrogel membranes can only collect water droplets in fog through their surface. They are not able to condense or absorb gaseous water in an environment with relative humidity from 50 to 90%.

We quantified the effect of conical geometries on water droplet transport and fog collection rate by fabricating and testing similar PVA/PPy gel membranes that contained equivalently spaced, geometrically identical surface micro-topologies of cones and cylinders, as well as flat surfaces. Figure 3c summarizes these findings and reveals that the micro-tree array exhibits a 34% higher fog collection rate than that of a flat surface, the cone array is 17% more efficient, and the cylinder array is 29% lower, after being normalized by total surface area. Since the directions of cones were not a key factor in the directional movement of the water drops[30], the effect of conical geometries was further studied by conducting systematic experiments on gel cone arrays with different conicity (Fig. 3d), which demonstrated that lower apex angles resulted in faster water collection rates; for example, the

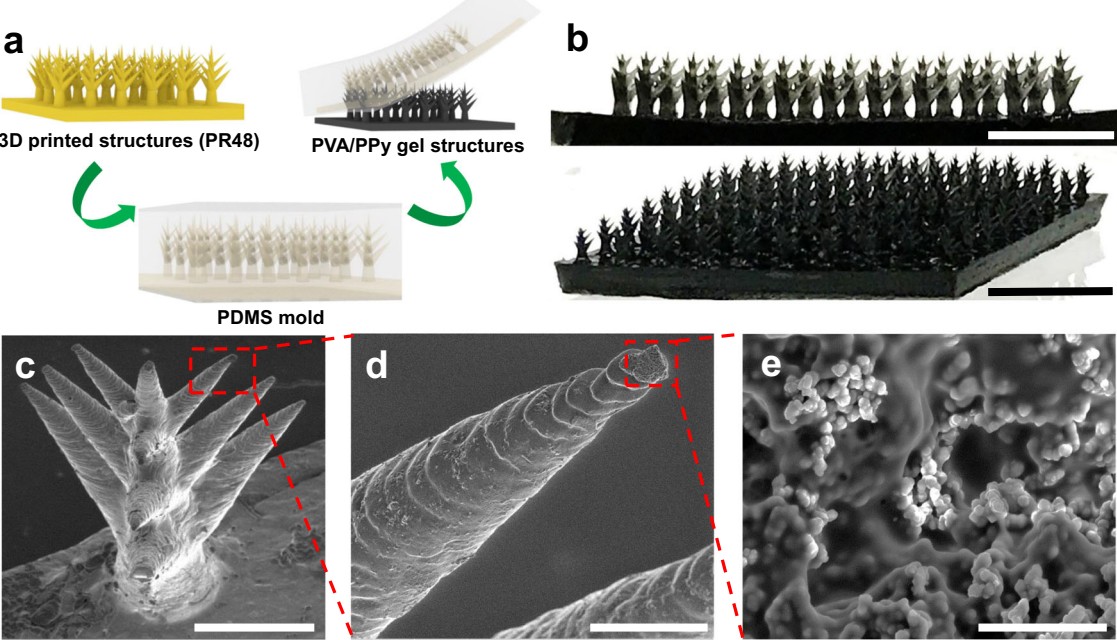

**Fig. 2 Fabrication and structure characterization of microstructured PVA/PPy gel membranes. a** Schematic illustration of the fabrication of microstructured PVA/PPy gel membranes. **b** Images of representative fabricated PVA/PPy gel micro-tree array. Scale bar: 1 cm. **c** Images of an individual representative tree micro-topology. Scale bar: 1 mm. **d** SEM image of one gel branch. Scale bar: 50 μm. **e** Porous structure of gel matrix. Scale bar: 5 μm.

collection rate (normalized by total surface area) of cone array increased by 14.7% when sinα decreased from 0.24 to 0.10 (Supplementary Fig. 7).

Figure 3e compares fog collection rates for several representative polymer meshes, as well as a Cactus spine and reveals that the areal efficiency of the PVA/PPy gel micro-tree arrays is 115% higher than that of double-layered Raschel mesh and 61% higher than that of a Cactus stem. The PVA/PPy gel micro-tree arrays also show the highest fog collection rates among all different materials based on the mass of polymeric materials (Supplementary Fig. 8). We evaluated the long-term stability and durability of micro-tree membranes by testing their structural integrity (Supplementary Fig. 4b, c) and bi-functional water harvesting properties for a twenty-month period in the lab. Figure 3f shows that the average fog collection rate, as well as solar vapor generation rate of PVA/PPy gel micro-tree array was well maintained after more than twenty-month storage.

Fog deposition and droplet transport are key processes that determine the fog collection performance[37,41]. Nucleation of water vapor and small water droplets is energetically more favorable on hydrophilic surfaces than hydrophobic ones[30,42,43]. An ideal fog collection structure should provide the enhanced surface area with hydrophilic nature to maximize droplet nucleation density[38]. Our design of micro-tree arrays is such that its footprint area of 1 cm$^2$ corresponds to a total surface area of ~3.5 cm$^2$ and increases the density of active sites for fog capture and droplet nucleation by increasing surface area (Supplementary Fig. 9– 11). The contact angle of 65° revealed the surface of PVA/PPy gel to be hydrophilic. As a comparison, membranes with the same geometric features printed out of PR48 (a commercial photo-resin) were hydrophobic, with a contact angle of 128°, and had a >65% lower fog collection rate (Supplementary Fig. 12). Membranes of pure PVA showed a similar contact angle to PVA/PPy gel and exhibited similar fog collection behaviors (Supplementary Fig. 13).

The conical structure of the PVA/PPy gel micro-trees enables efficient directional transport of deposited droplets, thus re-exposing the gel surface to incident vapor and accelerating the collection cycle. We compared fog collection behaviors of gel membranes with different surface topologies. Directional droplet movement was observed on gel cones (Supplementary Fig. 14) and the movement was faster as the apex angle decreased. On a tilted flat surface, initial water droplets randomly deposited and then increased their size through capturing drops in fog or coalescing with adjacent droplets but without obvious transfer of mass center (Supplementary Fig. 15). On gel cylinders, the droplet grew slowly while sticking on the cylinder until it fell (Supplementary Fig. 16). Both of these geometries do not lend themselves to quick regeneration of available droplet attachment which reduces the collection rate. Assembled by cones with the smallest apex angle (sinα = 0.10), our gel micro-trees array achieves the most efficient fog collection.

In addition, the hierarchical array provides a drag force resisting fog flow by lowering their speed in the region between the trees, thus increasing the possibility of droplets deposition on gel surface (Supplementary Fig. 17)[36]. This is also indicated by the varied time for different gel structures to reach their saturated collection rates, as shown in Fig. 3b. The flat membrane reached its maximum collection rate in the first 15 min because its whole surface was contacting with droplets right after it was exposed to fog flow while the gel micro-tree array showed much longer ramp time due to reduced flow speed and increased surface area. This dragging effect also affects the drainage of collected water and thus the size of the gel-tree array is tuned to facilitate the drainage (Supplementary Fig. 18).

**Solar steam generation by PVA/PPy gel membrane with micro-tree array.** PVA/PPy gel has been reported to be a highly efficient interfacial solar steam generator because it efficiently transports water through porous gel matrix and reduces water evaporation enthalpy[16]. We measured the solar steam generation properties of PVA/PPy gel membranes with different surface microstructures under 1 sun illumination (1 kW m$^{-2}$) by recording the overall mass change over 1 h, which represents the amount of evaporated

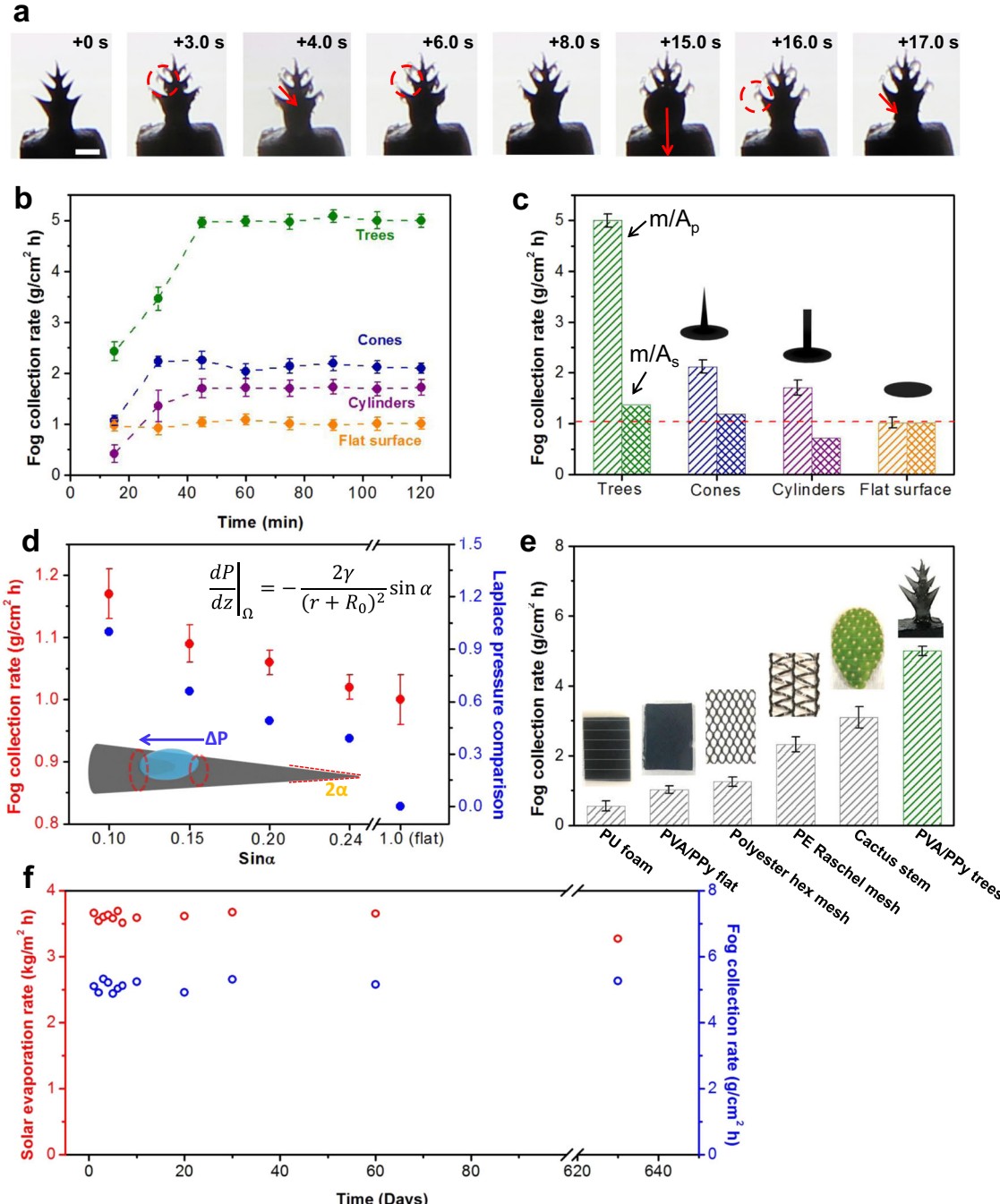

**Fig. 3 Fog collection properties of PVA/PPy gel membranes with micro-tree topologies. a** Snapshots of fog collection process for a single gel tree. Red circles correspond to droplet formation events and arrows point to droplet motion trajectories. Scale bar: 0.5 mm. **b** Fog collection rates measured for different gel membrane geometries as a function of time that demonstrates saturation at a particular time, unique to each geometry. **c** Fog collection rates of different gel membranes at steady states normalized by the projected area, $A_p$, (left column) and by the total surface area, $A_s$, (right column). **d** Fog collection rate and Laplace pressure difference as a function of apex angle. **e** Comparison to commercial meshes and a real Cactus stem. **f** PVA/PPy gel micro-tree array maintains dual water harvesting functions after more than twenty-month storage.

water. The membrane was floated on water and placed under the light beam. The mass of the water loss was measured every 10 min after the temperature of gel membranes achieved steady status. The PVA/PPy gel membrane with micro-tree array showed the best evaporation rate calculated per projected (illuminated) area, $A_p$, of 3.64 kg m$^{-2}$ h$^{-1}$, which is 7.1 times higher than that of free water and 14.1% higher than that of gel flat membrane (Fig. 4a). We fabricated additional hybrid gel micro-trees arrays with large areas (Supplementary Fig. 19) or with a

4 mm thick supporting layer and found that the water evaporation rate remained similar.

Energy efficiencies of different gel membranes can be calculated using[16]:

$$\eta = \dot{m} h_V / C_{opt} P_0 \qquad (2)$$

where $\dot{m}$ is the mass flux of evaporated water, h$_V$ is the vaporization enthalpy of the water, $P_0$ is the solar irradiation power (1 kW m$^{-2}$), and $C_{opt}$ is the optical concentration on

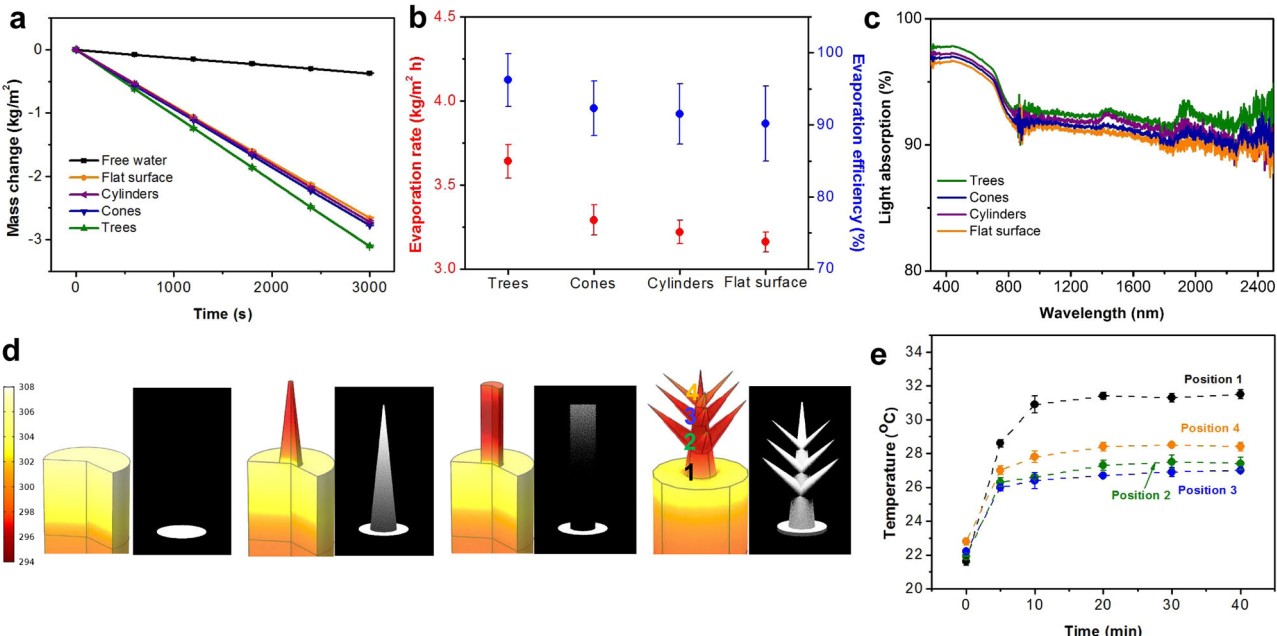

**Fig. 4 Solar steam generation properties of microstructured PVA/PPy gel membranes. a** Water loss for different membrane shapes under 1 sun, with free water as control. **b** Evaporation rate and energy efficiency for different tested micro-topologies. **c** Light absorption spectra over wavelengths of 250–2500 nm of gel membranes with different micro-topologies. The small jump of the curves at wavelength ~900 nm is caused by the switch of detectors. **d** Surface temperature contours (left) and illuminated pattern (right) for different micro-topologies under normal 1 sun illumination, simulated using COMSOL. **e** Measured surface temperature as a function of time at four positions along the height of a typical gel micro-tree.

absorber surface. Note that the water confined in hydrogel molecular mesh is evaporated to a state with a lower enthalpy change than conventional latent heat[16,18,44,45]. We used Raman spectra to confirm the existence of water molecules with different bonding states in PVA hydrogel and conducted controlled evaporation and differential scanning calorimetric (DSC) measurements to measure the evaporation enthalpy (Supplementary Figs. 20, 21). All gel membranes showed similar water evaporation enthalpy, which was demonstrated to be unaffected by micro-scale structures. The energy efficiencies of different gel structures are shown in Fig. 4b together with their evaporation rates, which conveys that PVA/PPy gel micro-tree array has the highest energy efficiency out of all tested geometries, and reaches up to ~96%, a factor of 65% greater than that of a porous plasmonic absorber[46] and 10% higher than that of carbon foam[12]. The PVA/PPy gel micro-trees array also shows the ability to effectively purify brines with different salt concentrations (Supplementary Fig. 22) and it will not contaminate the collected fog water (Supplementary Table 1).

The difference in water loss rates among gel membranes with different micro-topologies indicates that surface features, i.e., surface area, specific geometries, etc., affect solar steam generation. To understand the mechanisms, we examined the energy flow at steady-state by calculating the energy balance between solar irradiation, convection, radiation loss, evaporation, and loss to the water (Supplementary Fig. 23). We identified four structure-related factors that most significantly influence the energy flow: (1) light absorption, (2) surface area, (3) surface temperature, and (4) local humidity. We found that all gel membranes exhibited light absorption above 90% (Fig. 4c), with the micro-tree array having the highest absorption from wavelength of 250–2500 nm, possibly enhanced by increased light scattering within the "forest".

In an interfacial solar steam generator, the light-to-thermal energy conversion and water evaporation processes are confined to the gel-air interface, which implies that a large surface area and

a high equilibrium surface temperature are beneficial for steam generation. These two factors are found to be affected by surface microstructures due to structural shadowing and changed light incident angle[15,47,48]. We simulated and experimentally confirmed the temperature distribution within the PVA/PPy gel membranes subjected to normal incidence irradiation from the light in the solar simulator (Supplementary Fig. 23). The contour plots in Fig. 4d and the temperature vs. time plots at four different positions along the height of a representative tree shown in Fig. 4e, indicate that all gel microstructures have a lower average surface temperature at steady states compared with a flat surface. It appears that the cone absorbs light along its entire surface, thus reaching an average surface temperature of ~27.5 °C. The cylinder absorbs light only at the top surface, which results in a ~1.0 °C lower average surface temperature in the cylinder array and limits its overall water evaporation rate despite having a larger surface area compared with the cone array. By assembling the cones in a branched way, gel micro-trees can directly absorb sunlight through most of their surfaces, which enables maintaining a high average surface temperature of ~28.0 °C under 1 sun irradiation and results in energy efficiency close to 100%. We also noted that compared to flat membrane, microstructured gels allowed more heat consumption through increased gel-air interface and minimized the energy dissipated to the gel underneath the membrane surface, thus improving their energy efficiencies.

Local humidity near the gel-air interface also influences vapor generation and can be affected by surface morphology[16]. An ideal surface structure should facilitate easy escape of generated vapor since accumulated vapor leads to increased local humidity and hinders water evaporation. Assouline et al.[49], reported that an individual cone has a lower resistance to vapor flow compared with cylinder or inverted cone structures due to its convergent flow lines towards the narrow opening. In an array, the vapor flow is also affected by the eddy currents from the adjacent structures. In our experiments where the pitch among the features was systematically varied while other parameters remained fixed,

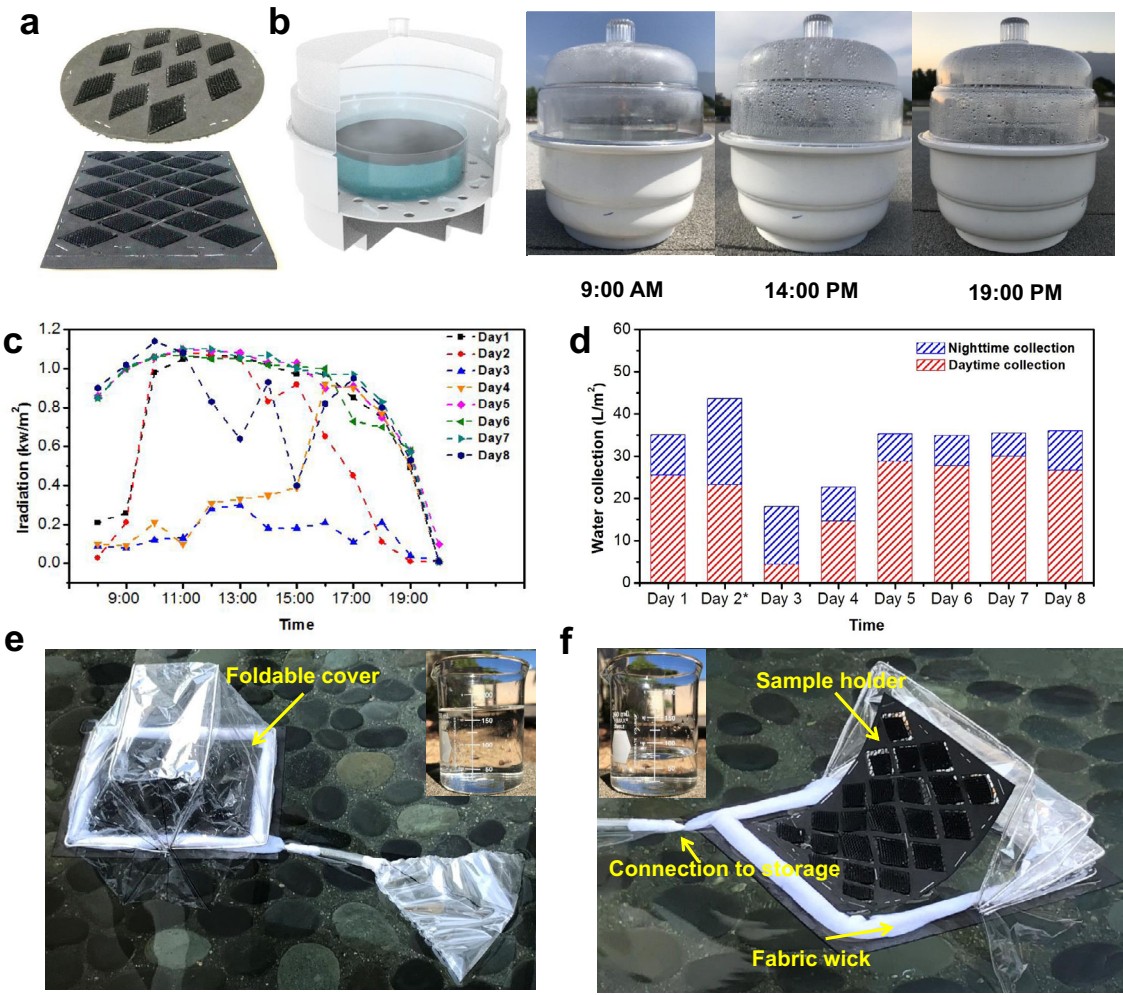

**Fig. 5 All-day water harvesting by PVA/PPy gel micro-tree array outdoors. a** Gel samples are held by a supporting structure made of polyurethane foam. **b** Schematic illustration and photos of a rooftop prototype acting as a solar water desalination system during daytime. **c** Solar radiation recorded during rooftop tests by portable solar power radiation meter. **d** Daily water collection per square meter of gel membrane during rooftop tests. Red: water collected during daytime (8 am–8 pm); blue: water collected during nighttime (8 pm to next day's 8 am). **e** Daytime and **f** nighttime modes of a floating water harvesting prototype. Insets show the water collected during a day (~170 mL) and night (~70 mL).

the gel cylinder array exhibited a performance drop of 15.5% when the cylinder distance was reduced by two times; the cone array maintained its performance as virtually unchanged over the separation distances of 0.4–1.2 mm (Supplementary Fig. 24a). We explain this behavior by the fact that the closely packed cylinders trap vapor more effectively compared with cone array. We introduce a shape factor as a figure of merit to qualitatively account for this effect (Supplementary Fig. 24b). We further compared two geometrical factors, shape factor and total surface area, of gel micro-trees array to those of other micro-structure arrays at different inter-distances (Supplementary Fig. 25). The results showed that the cone-based gel micro-trees array enables a larger evaporation area for a comparable shape factor, indicating that the generated vapor could still efficiently escape when the evaporation area is greatly increased.

**Efficiency and performance of all-day water collection by micro-topological PVA/PPy gel membranes**. We conducted a rooftop test to evaluate the water collection ability of micro-topological membranes in a natural environment by harvesting fog over 12 h periods, from 20:00 pm to the next day's 8:00 am and desalinating brine water under sunlight from 8:00 am to 20:00 pm (Supplementary Figs. 26, 27 and Supplementary

Table 2). The gel samples with total membrane areas of 55–126 cm$^2$ were held by a supporting structure shown in Fig. 5a and placed in a prototype device for all-day water collection (Fig. 5b). Solar irradiation was carefully traced every hour using a portable solar power radiation meter (Fig. 5c). The results in Fig. 5d showed that on a typical sunny day in Pasadena, CA, with an average solar heat flux of ~1 kW m$^{-2}$, the amount of collected water during daytime was ~150 mL and ~35 mL during nighttime, which translates into efficiencies of ~28 L m$^{-2}$ and ~6 L m$^{-2}$, correspondingly, based on the area of gel membranes. The water collection rates based on the total water surface area were also calculated to show the overall efficiency of our rooftop prototype (Supplementary Fig. 28). The average energy efficiency of microstructured gel membranes in the system is around 50%. Parameters including temperatures, wind speed, relative humidity of outdoor system were also recorded and analyzed for two daytime tests, which indicated that the restrained water vaporization was mainly caused by lower sunlight input and saturated internal humidity of the closed system (Supplementary Figs. 29, 30)[13]. It should be noted that the fog collection rates vary with the weather conditions (Supplementary Fig. 31). On cloudy nights, like those on days 1 and 3, around 10 L m$^{-2}$ of fresh water can be harvested. We expect this rate could be higher in a foggy location.

To demonstrate the versatility of our device, we also fabricated a floating all-day water collection prototype with a foldable cover, tunable supporting structure, and complete water storage components (Fig. 5e, f and Supplementary Fig. 32). During daytime, the condensation structure is closed and the generated vapor re-condenses on it, which is collected by fabric wicks and transported to water storage[50]. During night, the cover remains open, and the gel samples face the fog flow. We tested this prototype in a garden pool and managed to successfully harvest ~240 mL of fresh water in one day from ~126 cm$^2$ of micro-topological gel membranes. It's worthy to point out that fog is collected through the top surface of our hydrogel and is then directly transported to the clean water container.

## Discussion

We designed and demonstrated an all-day fresh water harvesting device that combines fog collection and interfacial solar steam generation enabled by 3D micro-topologies engineered on the surface of PVA/PPy hydrogel membrane. We utilized bio-inspired design principles and other design considerations for efficient fog capture, directional droplet transport, and quick drainage to guide our design and fabrication of 3D micro-tree arrays comprised of self-similar branched micro-cones that mimic the structure of spines on Cactus stem. We measured the fog collection rate of 5.0 g cm$^{-2}$ h$^{-1}$, which is 115% higher than that of commercially used Raschel mesh and 61% higher than that of a Cactus stem. Our results extend the materials pool to include highly hydrophilic hydrogel materials for the development of high-performance fog harvesting devices. Our work also shows the great potential of the hydrogel 3D printing technique for environmental applications.

We found that the presence of surface microstructures also improved solar steam generation of hydrogel membrane by providing a large surface area for thermal conversion and water evaporation. Beyond surface area, this work revealed that the specific micro-topologies influence equilibrium surface temperature and local humidity: for example, cylinder arrays exhibited lower surface temperature and inhibited vapor escape than conical ones with otherwise equivalent parameters. Conical structures were demonstrated here to be promising candidates for their abilities to increase light absorption area and reduce resistance for vapor flow. These factors, as well as other parameters like light scattering in gel array, distribution of heat on surface structure, and fabrication defects should be simultaneously accounted for to attain optimal steam generation performance. Our findings point towards novel strategies to improve interfacial solar steam systems by surface structure engineering to achieve high surface area with managed thermal energy distribution and local humidity. This can be a general strategy to stimulate the potential of other interfacial solar steam generation materials, especially those newly developed hydrogel materials with high hydratable and light-absorbing abilities.

Utilizing these multiple objectives and design principles, we demonstrate that a prototype PVA/PPy gel membrane with 4 mm-tall, 0.8 mm diameter, 1.2 mm-separated micro-tree array achieves a high solar water evaporation rate of 3.64 kg m$^{-2}$ h$^{-1}$ under 1 sun. Durability studies in a lab setting demonstrated that the membranes were stable and did not degrade in performance over the course of 20 months. These microstructured hydrogel membranes also derive their advantages from being bi-functional: when tested in a rooftop prototype over 8 days, a 55 cm$^2$ membrane produced ~185 mL of fresh water through solar evaporation under natural sunlight and via fog capture during nighttime (between hours of 8 pm and 8 am). Our results demonstrate the capability of micro-topological PVA/PPy hydrogel membranes as efficient 24-h water harvesters,

attaining daily water collection efficiencies of ~34 L m$^{-2}$, which—together with their straightforward fabrication methodology that is compatible with large-scale manufacturing renders them promising for practical water collection devices.

Besides the rooftop prototype, we showed a simple hand-made prototype to prove the concept of a floating water collection device with day and night working modes. It demonstrated that the solar steam generation and fog harvesting functions could be coupled in one floating system. The floating device can be improved in many ways to realize the full potential of our gel membranes, as well as other high-performance interfacial solar evaporators. For examples, a semi-spherical cover with high sunlight transparency can be installed. Other opening and closing mechanisms can be applied to the cover. Remote control functions can be used to remotely switch the working modes. A cooling part can be equipped to promote condensation. These works require efforts from optical, mechanical, thermal, electrical, and electronic engineering and can be very interesting topics for future studies.

## Methods

**Synthesis of PPy**. All chemicals are purchased from Sigma Aldrich and used as received. For a typical synthesis of PPy nanoparticles, 0.228 g of APS is dissolved in 10 mL deionized (DI) water and 0.069 mL pyrrole is mixed with 10 mL DI water by sonication for 10 min. APS and pyrrole solutions are then dropwisely added into 50 mL 1.2 M HCl aqueous solution with stir. After polymerization for 5 min, PPy solids are collected after washing and filtration with DI water and IPA. The purified PPy is then re-dispersed in DI water by sonication to form PPy solution.

**Preparation of PVA/PPy gel precursor solution**. PVA has a molecular weight of 15,000 g/mol and hydrolysis degree of ~97%. To prepare PVA/PPy gel precursor solution, 10 mL of 10 wt% PVA solution is mixed with 1 mL of 10 wt% PPy solution by sonication for 5 min. Then 200 μL of 25 wt% glutaraldehyde solution and 100 μL of 2 M HCl aqueous solution are added, followed by sonication for 5 min. The precursor solution is readily used to fabricate structured PVA/PPy gel membranes.

**Synthesis of PVA/PPy gels**. The as-prepared PVA/PPy gel precursor solution is filled into the PDMS mold with vacuum-assisted method and gelation is carried out for 2 h at room temperature. The obtained PVA/PPy hybrid gel is purified by immersing into DI water overnight. The formed PVA/PPy gel structures are properly taken out of the mold after being completely frozen and sent for multiple cycles of freeze-thaw processing. In each cycle, the gel was frozen at −20 °C for 2 h and then thawed in 30 °C water bath.

**Fabrication of microstructured PVA/PPy gel membranes**. To fabricate micro-structured PVA/PPy gel membranes, a double-inverse molding method is applied. The CAD-designed structure is firstly printed using a stereolithography 3D Printer (Ember, Autodesk) with a commercial resin (PR48). Then an inverse Poly-dimethylsiloxane (PDMS, Sylgard 184, Dow Corning) mold is prepared using the cured PR48 structures. Note that the PDMS mold can be repeatedly used, thus enabling large-quantity fabrication of PVA/PPy gel structures. The as-prepared PVA/PPy gel precursor solution is filled into the PDMS mold with vacuum-assisted method and after gelation, the formed PVA/PPy gel structures were properly taken out of the mold after being completely frozen.

**Materials characterizations**. The pictures and video of hydrogel samples and their fog collection behavior were recorded by a digital camera (Canon, 60D). The morphology and microstructure of samples were observed by Scanning Electron Microscopy (FEI, Versa 3D DualBeam) operating at 5 kV. Before observation, the hydrogels were freeze dried for 24 h. The Raman spectra of hydrogels and pure water were recorded by the Raman Spectrometer (Reinishaw, M100). During testing, hydrogel sample was placed on a glass substrate and water on the surface was removed by Kimswipe. Pure water was sandwiched by two glass substrates. The light absorption spectra of hydrogel samples were recorded by UV–vis–NIR spectrometer (Cary, UV/Vis/NIR 5000) equipped with an integrating sphere. The contact angle tests were conducted on a contact angle goniometer (RemaHart, Model 210). The FTIR spectra of hydrogels were recorded by the Fourier Trans-form Infrared Spectrometer (Thermo Mattson, Infinity Gold FTIR). Rheological experiments were performed by using a rheometer (TA instrument, AR 2000EX) in a frequency sweep mode.

**Fog collection test in lab**. To test the fog collection ability of microstructured PVA/PPy gels, a hydrogel membrane sample with 4 cm$^2$ projected area is placed

with a inclined angle of 45 degree to the horizontal surface, meanwhile, a sustained fog flow generated by ultrasonic humidifier (Levoit, LV600HH) with a velocity of about $1 \, m \, s^{-1}$ is kept blowing to the surface with a tilted angle (15 degree) to the tangent direction of the membrane at room temperature. The outlet of fog is kept 3 cm from the bottom of the gel membrane. The fog flow is just blown to the structured region and higher than the solid substrate, which helps avoid edge effect on the supporting layer. A beaker is placed under the gel sample to collect drained water and the amount of collected water is measured every 15 minutes. The room temperature for fog collection tests is 25 °C and the relative humidity in artificial fog flow is 100%.

**Solar steam generation test in lab**. Water evaporation experiments were conducted using a home-made solar simulator with 1 sun solar flux (AM1.5, 100 mW cm$^{-2}$). The room temperature for the solar steam generation test is 25 °C and the relative humidity is ~50%. The intensity of light was calibrated using a photodiode. The membrane was floated on water and placed under the light beam. After 10 min of pre-irradiation, the temperature of gel membranes achieved steady state and the mass of the water loss was then measured every 10 min. The dark condition-evaporation rate was also measured and used to calibrate vapor generation data. At least five samples were tested for each kind of gel membrane.

**Stability test of bi-functional PVA/PPy gel micro-tree array**. The bi-functional water harvesting properties of PVA/PPy gel micro-tree array were tested during long-term storage. The sample was exposed to artificial fog flow for 2 h first and the amount of collected water was recorded every 15 min. After fog collection test, the same sample was left in open space for 0.5 h and then tested for evaporation performance under 1 Sun light irradiation for 1 h to calculate its solar vapor generation rate.

## Data availability

The data that support the findings of this study are available from the corresponding authors upon request.

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

## Acknowledgements

We thank Dr. Bruce S. Brunschwig for help with the solar simulator and Daryl Yee for help with 3D printing. We thank Shu Yan for help with drawing schemes. J.R.G. acknowledges the financial support of the Resnick Sustainability Institute and of the Caltech Space Solar Power Project. H.A.A. acknowledges financial support from the Joint Center for Artificial Photosynthesis, a Department of Energy (DOE) Energy Innovation Hub, supported through the Office of Science of the U.S. Department of Energy under Award Number DE-SC0004993. O. I. acknowledges support from the Caltech Space Solar Power Project and the 3M foundation.

## Author contributions

Y.S. and J.R.G. conceived the idea. Y.S. performed materials fabrication and characterization. O.I. performed the numerical simulations. Y.S., O.I., H.A.A., and J.R.G. analyzed the data, discussed the results, and wrote the manuscript.

## Competing interests

The authors declare no competing interests.
