## [Peer Review File · Nature Communications]

REVIEWER COMMENTS

Reviewer #1 (Remarks to the Author):

In this work, the authors provided an interesting and novel way to improve the working efficiency of interfacial solar steamers. They designed and fabricated hydrogel membranes with cone-based surface microstructures. The resulting materials show the unique combination of solar steam generation and fog harvesting abilities and achieved clean water collection around the clock. The experimental measurements presented in the manuscript demonstrated high performance of these microstructured hydrogel membranes for both functions. The mechanism studies in this work have been significantly improved after revisions. The concerns from previous referees have been well addressed and the working mechanisms are clearly explained in the current version. I believe that the effects of surface microstructures on surface temperature and vapor distributions revealed in this study are very useful and of great interest to other researchers in the field. Moreover, I am excited to see that the delicate hydrogel microstructures are realized through scalable 3D printers. This fact, together with the outdoor test results presented in this work, indicates the potential of this technology for practical applications. This revised manuscript can be accepted for publication in Nature Communications. A minor suggestion is for the authors to include in the introduction some closely relevant reviews on hydrogels for water harvesting technologies to reflect the state of the art, such as Chem. Rev. 120, 7642 (2020); Nature Rev. Mater. 5, 388 (2020); ACS Mater. Lett. 2, 671 (2020).

Reviewer #2 (Remarks to the Author):

The authors have addressed part of the questions raised by reviewers. However, the all-day water collection rate needs to be further calculated and justified carefully. I am not convinced of the details of experiments. The reasons are as follows:

- (1) In their reply, the collection rate was calculated based on the area of gel samples. However, I notice that they arrange several small samples rather than a single large one and the small samples are not tightly packed, as shown in Fig. 5a and Fig. S31. If the spacing of the arrangement is essential, it is more reasonable to calculate the collection rate based on the overall area. Or the single larger samples should be fabricated for performance testing, which will make the data more credible.
- (2) They reported that the water vaporization enthalpy is significantly reduced to 1000 kJ kg⁻¹, which is even 300 kJ kg⁻¹ lower than the enthalpy reported by Yu, G. H., et al. [Nat. Nanotech. 13, 489-495 (2018)] How to explain the further decrease of enthalpy? Whether it's the micro-trees morphology or the different pore size or something else? And the differential scanning calorimetric (DSC) measurement needs to be supplemented to prove the reduced evaporation enthalpy.
- (3) As shown in Fig. 5e, the condensation device is equipped with a transparent plastic cover and without forced heat transfer. This device results in high optical loss caused by condensates and limited heat transfer, which have adverse effects on condensation [Energy & Environmental Science 11, 1510-1519]. Especially, in their design, the middle part of covering is horizontal, which impede the collection of water and further result in higher optical loss.
- (4) I think the first question of reviewer #4 has not been addressed. The author thinks that the fog

collection processes only occur on the gel surface and will not wash out the contaminants in the gel. However, because of high concentrations of pollutants in the absorber, diffusion of pollutants into fog water will occur. The inductively coupled plasma spectroscopy of fog water should be supplemented.

Point-to-point Responses to Referees' comments

Manuscript Title: All-day Fresh Water Harvesting by Microstructured Hydrogel Membranes

Manuscript ID: NCOMMS-21-04497-T

Reviewer #1 (Remarks to the Author):

In this work, the authors provided an interesting and novel way to improve the working efficiency of interfacial solar steamers. They designed and fabricated hydrogel membranes with cone-based surface microstructures. The resulting materials show the unique combination of solar steam generation and fog harvesting abilities and achieved clean water collection around the clock. The experimental measurements presented in the manuscript demonstrated high performance of these microstructured hydrogel membranes for both functions. The mechanism studies in this work have been significantly improved after revisions. The concerns from previous referees have been well addressed and the working mechanisms are clearly explained in the current version. I believe that the effects of surface microstructures on surface temperature and vapor distributions revealed in this study are very useful and of great interest to other researchers in the field. Moreover, I am excited to see that the delicate hydrogel microstructures are realized through scalable 3D printers. This fact, together with the outdoor test results presented in this work, indicates the potential of this technology for practical applications. This revised manuscript can be accepted for publication in Nature Communications. A minor suggestion is for the authors to include in the introduction some closely relevant reviews on hydrogels for water harvesting technologies to reflect the state of the art, such as Chem. Rev. 120, 7642 (2020); Nature Rev. Mater. 5, 388 (2020); ACS Mater. Lett. 2, 671 (2020).

Response:

We thank the referee for the positive comments. We have cited these important references in the Introduction section.

Reviewer #2 (Remarks to the Author):

The authors have addressed part of the questions raised by reviewers. However, the all-day water collection rate needs to be further calculated and justified carefully. I am not convinced of the details of experiments. The reasons are as follows:

(1) In their reply, the collection rate was calculated based on the area of gel samples. However, I notice that they arrange several small samples rather than a single large one and the small samples are not tightly packed, as shown in Fig. 5a and Fig. S31. If the spacing of the arrangement is essential, it is more reasonable to calculate the collection rate based on the overall area. Or

the single larger samples should be fabricated for performance testing, which will make the data more credible.

Response:

We thank the referee for the constructive comments. Due to the limitation of the 3D printer, we are unable to fabricate a single large gel sample (more than 50 cm²) to fit in the rooftop tests. However, we had studied the fog collection and solar steam generation performance of gel membranes with different areas (Fig. S11 and Fig. S19). The results show that the area of membrane has little effects on either function, which indicates that the performance of gel membranes will not be affected whether they are tightly packed or not.

In the rooftop tests, we used a passive polyurethane foam, which has a water evaporation rate close to free water and a low fog collection rate, to tightly support and fix our gel samples. We also conducted a control experiment during rooftop tests to subtract the contribution from the PU foam. The processed results reflect the working performance of microstructured gel membranes in the rooftop tests, which is the focus of our study. This set-up also provides the flexibility for people to arrange gel membranes in different ways (Fig. 5a), thus fitting their water containers in house or rooftop devices made by themselves.

We also agree with the referee that the collection rate based on the total water surface area should be provided to show the overall efficiency of our rooftop prototype. The results are shown in Figure R1 below. We can see that the daily water collection rate is much lower because about half of the water surface was covered by the passive PU foam. As a comparison, the PU foam itself showed a daily water collection rate of 1.0 to 2.2 L/m² in our control experiment. The collection capability of rooftop device can be improved by assembling more PVA/PPy gel membranes. We have added the data in Section 22 of Supplementary Information.

Fig. R1 Daily water collection per square meter of total water surface during rooftop tests. Red: water collected during daytime (8 am to 8 pm); blue: water collected during nighttime (8 pm to next day's 8 am).

(2) They reported that the water vaporization enthalpy is significantly reduced to 1000 kJ kg⁻¹, which is even 300 kJ kg⁻¹ lower than the enthalpy reported by Yu, G. H., et al. [Nat. Nanotech. 13, 489-495 (2018)] How to explain the further decrease of enthalpy? Whether it's the micro-trees morphology or the different pore size or something else? And the differential scanning calorimetric (DSC) measurement needs to be supplemented to prove the reduced evaporation enthalpy.

Response:

We thank the referee for bringing up this important point. We actually had noticed this decrease of enthalpy and discussed with other researchers including the authors of Nat. Nanotech paper about this enthalpy difference. We found that this was most possibly caused by the different hydrolysis degree of PVA used in our study. The PVA reported in the Nat. Nanotech. paper had a hydrolysis degree of ~88% and the PVA used in our study had a hydrolysis degree of ~97% (the hydrolysis degree of our PVA had been specifically mentioned in the Experimental Section). PVA with a higher hydrolysis degree contains more –OH groups on their polymeric chains, which could bind more water molecules and reduce the enthalpy. Similar results have been found in other studies. In the Nat. Nanotech. paper¹, a control gel sample with a higher PVA concentration showed an enthalpy around 1000 kJ/kg. Zhou, X., et al.² reported that by adding small amounts of chitosan (which contains more –OH and –NH₂ groups than PVA), the water evaporation enthalpy of PVA/PPy gel system can be tuned from ~1400 to ~800 kJ/kg. Our measurements proved that the enthalpy was not affected by the surface microstructures (Fig. S21). Moreover, the porous microstructure of PVA/PPy gel was not affected by the hydrolysis degree of PVA since the molecular weight of PVA, concentrations of crosslinkers, and concentrations of PVA and PPy remained same.

As suggested by the referee, we used DSC to measure the water vaporization enthalpy in our gel, as shown in Figure R2 below. The gel sample was placed in an open Al crucible and measured with a linear heating rate of 5 K/min, under a nitrogen flow (20 mL/min), in the temperature range from 20 to 180 °C. The effective specific heat capacity was calculated by comparing the heat flow of measured gels with that of the standard sapphire sample. We firstly validated our measurements by conducting the tests on free water. The measured enthalpy of free water is 2424 kJ/kg, which is very close to the theoretical value of 2450 kJ/kg. The measured water vaporization enthalpy in our hybrid gel is 1735 kJ/kg. As comparisons, the gel sample reported in Nat. Nanotech. paper¹ has a DSC measured enthalpy of 1919 kJ/kg (which has an enthalpy of ~1300 kJ/kg in evaporation experiment) and its control sample with a higher PVA concentration has a measured enthalpy of 1765 kJ/kg. As mentioned above, this control sample also showed an enthalpy around 1000 kJ/kg in the evaporation experiment. Note that the enthalpy values calculated from DSC are higher than those tested in evaporation experiments, since the DSC test and evaporation test present a full dehydration and slightly dehydration processes, respectively¹. Our DSC measurements confirmed the results from our evaporation

tests. We have added the DSC results and related discussions in Section 15 of Supplementary Information.

We want to further stress on that the water evaporation rate of gel evaporators is determined by several factors including porous network, light absorption ability, water content, water vaporization enthalpy, working conditions (environmental temperature, humidity, etc.) surface structures, etc. Our study focused on the effects of surface microstructures and provided meaningful information about regulating the surface temperature and vapor flow through surface microstructure design.

Fig. R2 DSC measurements on free water and PVA/PPy hybrid gel.

(3) As shown in Fig. 5e, the condensation device is equipped with a transparent plastic cover and without forced heat transfer. This device results in high optical loss caused by condensates and limited heat transfer, which have adverse effects on condensation [Energy & Environmental Science 11, 1510-1519]. Especially, in their design, the middle part of covering is horizontal, which impede the collection of water and further result in higher optical loss.

Response:

We agree with the referee's comments on our floating prototype, which has a lower working efficiency (a daily water collection rate of $\sim 19 \text{ L/m}^2$) due to the problems pointed out by the referee. However, our study focused on the design of micro-structured gel membranes as interfacial solar steam generation materials and understanding their intrinsic properties, but not on the optimization of the whole water collection system. We showed this simple hand-made prototype to prove the concept of a floating water collection device with day and night working modes. It demonstrated that the solar steam generation and fog harvesting functions could be coupled in one water collection device.

We understand that the floating device can be improved in many ways to realize the full potential of our gel membranes, as well as other high-performance interfacial solar evaporators. For examples, a semi-spherical cover with high sunlight transparency can be installed. New opening and closing mechanisms can be applied to the cover. Remote control functions can be used to remotely switch the working modes. A cooling part can be equipped to promote the condensation. However, we realize that these works require efforts from optical, mechanical, thermal, electrical, and electronic engineering and thus are beyond the focus of this study. Development of water condensation and collection systems for large-scale applications can be a very interesting topic for future studies. We have added more perspectives on floating device improvement in the Discussion Section.

(4) I think the first question of reviewer #4 has not been addressed. The author thinks that the fog collection processes only occur on the gel surface and will not wash out the contaminants in the gel. However, because of high concentrations of pollutants in the absorber, diffusion of pollutants into fog water will occur. The inductively coupled plasma spectroscopy of fog water should be supplemented.

Response:

We thank the referee for the suggestion to assess the quality of collected fog water. We are sorry that we couldn't find a plasma spectroscopy for tests during the Covid-19 pandemic. To address the referee's concerns, we designed a control experiment in lab. To mimic a gel sample which is possibly contaminated during solar evaporation, we used a PVA/PPy gel sample to evaporate NaCl solution with a very high salinity of 100 under one Sun irradiation for 6 h. Then we conducted fog collection test on this gel sample in lab for 6 h (the set-up can be found in Section 4 of Supplementary Information). The artificial fog was generated by adding dilute NaCl solution into the humidifier (so we can measure the salinity by conductance tests, as described in Section 16 of Supplementary Information). The salinity of collected fog water was measured and compared to that of water in the humidifier. This solar evaporation and fog collection cycle was repeated for 10 times using the same gel sample. The salinity of water collected during each fog collection process is recorded in the Table below. We can see that even when we used highly concentrated saline water during solar evaporation, the salinity of water collected during fog collection was not increased. The results demonstrate that the collected fog water is not contaminated by the gel evaporator.

This can be explained by these reasons: 1. The hydrogel evaporators show an antifouling property and long-term stability. The salts or pollutants don't accumulate in the gel membrane but mainly remain in the original solution^{3,4}. 2. The fog collection process on our micro-structured gel membrane is fast. The cycle of fog droplets nucleation followed by their transport, growth, and eventual drainage of the large water drops takes an average period of ~20 s and the fog droplets keep moving on the surface of gel membrane (Fig. 3a, Fig. S6, and Movie S1). In such

short time, the diffusion of low-concentration pollutants in the gel matrix into the quickly moving fog droplets on the gel surface is difficult to occur. 3. In all fog collection processes (in lab, rooftop test, or in floating device), the gel membranes are tilted to facilitate the droplets transport (Fig. S5a, Fig. S26, and Fig. 5f) and they have no contact with the bottom saline water. Thus it's not possible for the pollutants to diffuse from the bottom water to the fog droplets on gel top surfaces through gel membranes.

We have added the experimental results and related discussions in Section 16 of Supplementary Information.

Table R1. The calculated salinity of water during in lab fog collection test.

Fog collection test #	Average salinity of collected water (‰)	Average salinity of water in humidifier (‰)
1	0.00126	0.00119
2	0.00148	0.00122
3	0.00137	0.00138
4	0.00119	0.00121
5	0.00122	0.00135
6	0.00130	0.00117
7	0.00134	0.00129
8	0.00125	0.00109
9	0.00113	0.00136
10	0.00117	0.00125

References

1. Zhao, F. et al. Highly efficient solar vapour generation via hierarchically nanostructured gels. *Nat. Nanotech.* 13, 489-495 (2018).
2. Zhou, X., Zhao, F., Guo, Y., Rosenberger, B. & Yu, G. Architecting highly hydratable polymer networks to tune the water state for solar water purification. *Sci. Adv.* 5, eaaw5484 (2019).
3. Zhou, X., Zhao, F., Guo, Y., Zhang, Y. & Yu, G. A hydrogel-based antifouling solar evaporator for highly efficient water desalination. *Energy Environ. Sci.* 11, 1985-1992 (2018).
4. Guo, Y. et al. Biomass-Derived Hybrid Hydrogel Evaporators for Cost-Effective Solar Water Purification. *Adv. Mater.* 32, 1907061 (2020).

REVIEWERS' COMMENTS

Reviewer #2 (Remarks to the Author):

All the issues have been addressed. It can now be published.